# A Simple Colorimetric and Fluorescent Sensor to Detect Organophosphate Pesticides Based on Adenosine Triphosphate-Modified Gold Nanoparticles

**DOI:** 10.3390/s18124302

**Published:** 2018-12-06

**Authors:** Xiaoxia Li, Haixin Cui, Zhanghua Zeng

**Affiliations:** Institute of Environment and Sustainable Development in Agriculture, Chinese Academy of Agricultural Sciences, Beijing 100081, China; lixiaoxia1292@gmail.com

**Keywords:** organophosphate pesticides, gold nanoparticles, colorimetric and fluorescent sensor, ethoprophos detection, multimodal assay

## Abstract

A simple and dual modal (colorimetric and fluorescent) sensor for organophosphate pesticides with high sensitivity and selectivity using adenosine triphosphate (ATP)- and rhodamine B-modified gold nanoparticles (RB-AuNPs), was successfully fabricated. This detection for ethoprophos afforded colorimetric and fluorescence imaging changes visualization. The quantitative determination was linearly proportional to the amounts of ethoprophos in the range of a micromolar scale (4.0–15.0 µM). The limit of detection for ethoprophos was as low as 37.0 nM at 3σ/*k*. Moreover, the extent application of this simple assay was successfully demonstrated in tap water samples with high reliability and applicability, indicating remarkable application in real samples.

## 1. Introduction

Pesticides have been widely used worldwide to protect crops from insects and increase the quantity and quality of agricultural products [1,2,3]. Pesticide use, however, can affect ecosystems, nontarget organisms, agricultural product safety, and human health [4]. Organophosphates (OPs) are a class of pesticides designed to affect the insect nervous system [5,6,7,8,9,10]. OPs irreversibly inactivate acetylcholinesterase, an enzyme that degrades the neurotransmitter acetylcholine at cholinergic synapses and that is essential to nerve function [11,12]. OP residues in agricultural products and water are a potential source of toxicity to bees, wildlife, and humans [13,14]. It is therefore imperative to develop highly sensitive and reliable methods to determine pesticide content in environmental samples.

Multiple methods to assay OPs have been developed, including gas chromatography-mass spectrometry, liquid chromatography, enzyme inhibition, enzyme-linked immunosorbent assay (ELISA), and electrochemical analysis [15,16,17,18,19,20,21]. Chromatography is typically used for assaying pesticides, and liquid or gas chromatography-mass spectrometry is the most common method. Whereas chromatography methods provide adequate selectivity and sensitivity, they require multistep sample pretreatment, detailed experimental conditions, and adept operators and sophisticated instruments. The enzyme inhibition method is a fast and simple low-cost technique, but its specificity is low. ELISA uses antibodies and color changes to identify a substance with good repeatability, sensitivity, and specificity, but the poor stability of enzymes or antibodies in practical performance and the low specificity limit its use. None of the conventional methods are suitable for on-site detection with sensitivity and specificity. Significant challenges remain in exploiting fast, specific, low-cost, and easy-to-manipulate OP assays.

Gold nanoparticles (AuNPs) possess unique optical properties related to their size, surrounding environment, and dispersion state [22,23]. AuNP-based colorimetric and fluorescent assays have been used for chemical and biological applications with versatility and simplicity [24,25,26,27,28,29,30]. These assays are based on AuNP dispersion and aggregation states in response to a specific analyte and certain recognition mechanisms. The surface energy of AuNPs tends to decrease, leading to aggregation and sedimentation of the colloidal system. Citric acid-modified AuNPs can be easily aggregated by the addition of sulfur-containing OPs because of the strong Au-S bond. In recent years, this AuNP aggregation mechanism has drawn considerable attention in OP research [31,32,33,34]. Although these methods are accurate, fast, sensitive, and easy to perform, they commonly lack specificity to some OPs because of the sensitive, strong Au-S bond, as many OPs contain sulfur. In theory, electrostatic and steric repulsion are the two main stabilizing mechanisms against AuNP aggregation, and the strong Au-S bond can destroy AuNP electrostatic stabilization and result in aggregation, depending on the OPs’ sulfur energy and capping agents. In this paper, we discuss the possibility of identifying and quantifying specific sulfur-containing OPs by controlling the Au-S bond and capping agent. We established a specific dual-readout assay for the OP ethoprophos by adding adenosine triphosphate (ATP) to AuNPs and controlling the ratio of ATP to AuNPs. This method was validated in tap water sample analysis.

## 2. Materials and Methods

### 2.1. Chemicals and Instruments

Chlorpyrifos (CP), ethoprophos (EP), profenofos (PF), trichlorfon (TC), omethoate (OT), monocrotophos (MC), isocarbophos (IC), and malathion (MT) were purchased from Honeywell Fluka (Morris Plains, NJ, USA). Adenosine 5′-triphosphate disodium salt hydrate (ATP) was obtained from Sigma-Aldrich (St. Louis, MO, USA). Gold(III) chloride trihydrate (49.0% Au) and sodium citrate tribasic dihydrate were purchased from J&K Scientific (Beijing, China). All commercial chemicals were used directly without any purification.

The color changes caused by AuNP aggregation were measured by ultraviolet-visible light spectroscopy absorption (400–900 nm) using a Shimadzu 2600 spectrophotometer (Shimadzu Corp., Kyoto, Japan). Because the AuNP solution was highly concentrated, we double-diluted the solution and set the corresponding absorption band to 520 nm. The fluorescence intensity of the spectra was recorded at a range of 540–700 nm, operating at an excitation wavelength of 520 nm using a Hitachi F7000 fluorescence spectrophotometer (Hitachi, Tokyo, Japan). The weak fluorescence intensity of rhodamine B-AuNP (RB-AuNP) solutions at approximately 565 nm indicated that the fluorescence of the rhodamine B molecules was completely quenched by the AuNPs, confirming that nearly all rhodamine B molecules were adsorbed onto the surfaces of the AuNPs. The excitation and emission slit widths were both 10 nm. Fluorescence intensities were captured using an inverted fluorescence microscope (Olympus, Tokyo, Japan). The diameter and morphology of AuNPs were imaged by transmission electron microscope (Hitachi HT7700), and hydrodynamic size distribution was measured using a particle size analyzer (Nano-ES90, Malvern Instruments, Malvern, UK).

### 2.2. Preparation of Stock Solutions

Stock solutions of the pesticides (10.0 mM) were prepared in absolute ethanol and used within one month. The pesticide stock solutions were diluted 10-fold with absolute ethanol, to obtain 1.0 mM solutions.

The standard serial solution of ethoprophos was one milliliter of 1.0 mM ethoprophos solution transferred to a 50-mL volumetric flask and diluted with deionized water to a specific volume, then fully mixed. The final concentration was 20.0 μM. Aliquots (1.0, 2.0, 3.0, 4.0, 5.0, and 7.5 mL) of the pre-prepared ethoprophos (20.0 μM) solution were transferred to 10.0-mL volumetric flasks, diluted with deionized water to a specific volume, and fully mixed. The corresponding concentrations of these serial solutions were 2.0, 4.0, 6.0, 8.0, 10.0, and 15.0 μM.

ATP (10.0 mM) and rhodamine B (10.0 mM) were prepared in deionized water and used within one week. For lower concentrations of solutions used in later experiments, the ATP and rhodamine B stock solutions were diluted with deionized water and used within 8 h.

### 2.3. Preparation of AuNPs

The AuNPs used in this experiment were water-soluble and had diameters of approximately 13 nm, as determined by transmission electron microscopy. The sodium citrate solution preparation described by Hauser and Lynn was used to synthesize citrate-capped hydrophilic AuNPs [35]. By varying the citrate/Au^3+^ molar ratio according to the modification by Turkevich and Frens [36], the diameters of the AuNPs were tuned to a range of 10–150 nm.

### 2.4. Citrate-Capped Hydrophilic AuNPs

HAuCl_4_ (40.2 mg, 49.0% Au) was dissolved in 100.0 mL deionized water and heated to boiling. The HAuCl_4_ solution appeared yellow. Ten milliliters of trisodium citrate solution (38.8 mmol/L) were then added slowly to the HAuCl_4_ solution with stirring. Color changes accompanied the reaction, from yellow to dark blue to a final transparent wine-red. The solution was heated for another 10.0 min. After cooling to room temperature, the obtained aqueous citrate-capped AuNP solution was filtered through a 0.22-μm hydrophilic membrane to remove insoluble residues and large solid particles.

### 2.5. Preparation of ATP-Modified AuNPs and Rhodamine B-Coated AuNPs

AuNPs were protected by ATP before coating with rhodamine B. First, a stock solution of ATP (1.0 mM, 1.0 mL) was added to the citrate-capped hydrophilic AuNPs solutions (50.0 mL). The mixed solutions were vortexed for 30.0 min at room temperature. Next, 1.5 mL rhodamine B solutions (1.0 μM) were slowly added to the ATP-protected AuNPs solutions and vortexed lightly in the dark at room temperature for 30.0 min. The resulting RB-AuNPs solutions were stored at 4.0 °C as stock solutions for later use. The particle size of the RB-AuNPs was approximately 15 nm, and the particles were mono-dispersed in the solution.

### 2.6. Citrate-Capped AuNPs Pesticide Assay

Citrate-capped AuNPs were first double-diluted for this experiment. The pesticide solutions (50.0 μL, 1.0 mM) were added to the double-diluted citrate-capped AuNPs solutions (5.0 mL). Ethanol (50.0 μL) was added to the citrate-capped AuNPs solution (5.0 mL) as a blank, which eliminated interference and improved the precision of analysis. The blank test was repeated for 12 replicates to calculate the limit of detection (LOD). The mixtures were vortexed for 1.0 min and kept in the dark for 15.0 min before being imaged, and the ultraviolet-visible light absorption spectra (400–900 nm) were collected. All measurements were repeated for at least three replicates.

### 2.7. ATP-Protected AuNPs Pesticide Assay

The experimental procedure and conditions for the citrate-capped AuNPs assay were repeated, replacing citrate-capped AuNPs with RB-AuNPs. The solutions were imaged and the ultraviolet-visible light absorption spectra (400–900 nm) were collected. All measurements were repeated for at least three replicates.

### 2.8. Experimental Procedure for Ethoprophos Detection Sensitivity

The standard ethoprophos serial solutions (0.5–100.0 μM) were mixed with the RB-AuNPs in a ratio of 1:1 in ultramicro cells (Agilent Technologies, Santa Clara, CA, USA), vortexed for 20 s, and the ultraviolet-visible light absorption spectra (400–900 nm) were recorded every 1 min 30 times. The LOD was calculated as 3σ/*k*, where σ is the standard deviation of the 12 blank tests, and *k* is the slope of the linear fitting equation.

## 3. Results and Discussion

### 3.1. Aggregation-Inducing Effect of OPs on Citrate-Capped AuNPs

We first analyzed the aggregation-inducing effect of the OPs (Figure 1) on citrate-capped AuNPs (Figure 2a). We calculated the absorption change at 520 nm (Δ*A*_520_) by setting the blank value as zero. The corresponding LOD was also adjusted by deducting the blank value. We found that ethoprophos, profenofos, and chlorpyrifos induced aggregation of citrate-capped AuNPs. This was confirmed by the AuNPs’ color changes from red to purple and blue. The Δ*A*_520_ was 0.809 (ethoprophos, 10.0 μM), 0.399 (profenofos, 10.0 μM), and 0.129 (chlorpyrifos, 10.0 μM). The AuNP solutions did not change color in the presence of the other pesticides (1.0 mM) or ethanol. The Δ*A*_520_ for malathion could not be determined. The Δ*A*_520_ values of trichlorfon, omethoate, monocrotophos, isocarbophos, and malathion were determined to be much smaller than those of ethoprophos and profenofos (Figure 2). These findings indicated that although a visible aggregation of citrate-capped AuNPs was not observed, these four pesticides could cause a slight aggregation of citrate-capped AuNPs.

### 3.2. C.olorimetric and Fluorescent Detection of OPs by RB-AuNPs

To improve the detection specificity of AuNPs for OPs, we measured the effects of OPs on RB-AuNPs, which were capped with ATP on the surface. ATP has a greater affinity for AuNPs than citric acid, because both adenine and triphosphate show some binding to AuNPs. This binding affinity is ATP concentration-dependent. We envisioned that AuNPs capped with a specific concentration could show specific assays for OPs based on the displacement of ATP and OPs. To selectively assay for ethoprophos, an ATP concentration was optimized to be 1.2 mM (a detailed discussion is presented in the next section). As expected, only ethoprophos caused a color change in RB-AuNPs, from wine-red to a pink color within 15.0 min (Figure 3a), indicating improved specificity for ethoprophos compared to findings from citrate-capped AuNPs (Figure 3b). The absorption responses of RB-AuNPs to ethoprophos were concentration-dependent (Figure 3c). Ethoprophos yielded a relatively large positive Δ*A*_520_ of 0.205, and the color of the RB-AuNPs changed: The change was correlated with the concentration of ethoprophos (Figure 3c). The absorption at 520 nm decreased along with an increase at 600 nm, and the curve of the ratio of absorption at 600 nm and 520 nm (ROA_600/520_) versus ethoprophos concentration was plotted (Figure 3d). These results were indicative of RB-AuNPs’ aggregation, which was confirmed by transmission electron microscopy (Figure 4). Profenofos (10.0 μM) had a rather low Δ*A*_520_ of 0.028, without a color change. The Δ*A*_520_ values of the other six OPs pesticides (1.0 mM) were very small, and the color of the AuNPs did not change. In addition, the competition experiments were studied by measuring ROA_600/520_ in the presence of ethoprophos (4.0 μM) and interfering analytes with a specific concentration (4.0 μM). Very few changes in ROA_600/520_ were observed in the combination of ethoprophos and interfering pesticides, indicating that this simple assay for ethoprophos could be applicable in the complex samples.

It is well known that AuNPs quench fluorophore fluorescence almost completely. Here, rhodamine B molecules were tightly adsorbed to the surface of AuNPs via electrostatic attraction to ATP, which was confirmed by the negligible fluorescence intensity of the RB-AuNPs. The fluorescent results showed that RB-AuNPs had a strong, concentration-dependent fluorescence recovery response to ethoprophos (Figure 5) and a negligible response to the other OPs.

### 3.3. The Effect of ATP Concentration on OP Detection by RB-AuNPs

We found that after modification of the AuNPs with ATP, OP-induced aggregation of the NPs was inhibited to some extent. In the first experiment, using citrate-capped AuNPs, the OPs ethoprophos, profenofos, and chlorpyrifos caused notable aggregation of AuNPs, visualized by the color change from wine-red to dark blue after 15 min. Added ATP, however, replaced citric acid as the stabilizing agent, as ATP has a greater affinity to AuNPs [37,38,39]. RB-AuNPs were not induced to aggregate by profenofos or chlorpyrifos, but only by ethoprophos. The corresponding Δ*A*_520_ change in ultraviolet-visible light absorption declined markedly from 0.809 to 0.205, indicating that ATP enhanced AuNPs’ stability and reduced the tendency to aggregate. Citrate-capped AuNPs had slight absorption responses to the other pesticides, whereas RB-AuNPs were not responsive. These findings suggest that ATP is protective of AuNPs and reduces OP-induced aggregation.

ATP concentrations and the detection reaction time were determining factors in this assay. Changes in the absorption spectra and color of RB-AuNPs were negligible for chlorpyrifos, trichlorfon, omethoate, monocrotophos, isocarbophos, and malathion, even at a high concentration of 50.0 mM. In the case of profenofos, slight variations in the absorption spectra and color of the RB-AuNPs were observed at low ATP concentrations or short reaction times of detection, suggesting that it may interfere with detecting ethoprophos. By optimizing the amount of ATP, we further tested various concentration of ATP. The results showed that the response to ethoprophos was in the range of 0.8–1.4 mM. At low concentrations of ATP, the sensitivity of the assay for ethoprophos was high. However, the RB-AuNPs showed some response to profenofos. At high concentrations of ATP, the sensitivity of the assay for ethoprophos was low, as there was no response to profenofos. As a result, the RB-AuNPs showed some response to profenofos in the low concentrations, and it took a long time to assay ethoprophos in the high concentrations. Given specificity and quickness, 1.0–1.2 mM ATP was optimized as a final concentration range. This established a method for detecting ethoprophos exclusively. The detailed data are summarized in Table 1. For the reported AuNPs-based OP sensors (Table 2), most of them were based on the inactivation of acetylcholinesterase and presented a lack of selectivity. Compared with them, this assay exhibited much more specificity and simplicity in the detection of ethoprophos.

### 3.4. The pH Effect on the Colorimetric Assay for Ethoprophos

Usually, pH shows a great effect on AuNP-based colorimetric sensors. The pH effect on this assay was studied in detail. Figure 6 reveals the Δ*A*_520_ of RB-AuNPs with 10.0 μM ethoprophos at various pH values (from 3.0 to 10.0). The Δ*A*_520_ of RB-AuNPs in the presence of ethoprophos was stable (around 0.8) under acidic and weakly basic conditions. However, it dramatically changed to 0.04 at pH 10.0, probably due to the decomposition of ethoprophos under basic conditions. These results suggest that this assay is practical and applicable under broad pH conditions.

### 3.5. Mechanism for Ethoprophos Detection

Sulfur-containing OPs can replace citric acid and lead to aggregation of AuNPs because of a strong Au-S binding tendency. ATP has high affinity for AuNPs, and can be considered a good stabilizing agent: ATP molecules adsorbed to the surface of RB-AuNPs can compete with sulfur-containing OPs, depending on the ATP concentration. Therefore, a specific assay for detecting sulfur-containing OPs could be refined by ATP concentration. Ethoprophos has a great affinity for AuNPs because it contains two sulfur atoms and forms stronger Au-S bonds than other OPs. It can demolish the frameworks of RB-AuNPs to cause nanoparticles aggregation, affording colorimetric change and fluorescence recovery (Scheme 1). An assay for ethoprophos could therefore be established if the ATP concentration was appropriate.

### 3.6. Ethoprophos LOD in RB-AuNPs

We used various concentrations of ethoprophos (0–20.0 μM) to study the absorption spectra of RB-AuNPs, and found that the decrease of adsorption at 520 nm and the increase of adsorption at 600 nm correlated with ethoprophos concentration (Figure 7b). The variations in absorption spectra indicated the extent of RB-AuNPs’ aggregation. The colorimetric variation was observed in this range (Figure 7a). The ratio of the absorption value at 600 nm to the absorption value at 520 nm at various ethoprophos concentrations was plotted to calculate the LOD, which was determined to be 0.089 mg/L (37.0 nM) at 3σ/*k*. The quantitative range was 4.0–15.0 μM.

### 3.7. Ethoprophos Detection in Real Samples

Ethoprophos is commonly used for nematode control, and off-target spraying pollutes the surrounding environment and sources of drinking water. We used our RB-AuNPs assay to detect ethoprophos in tap water, anticipating that the complex matrices in a real sample would not interfere with accurate detection. A certain amount of ethoprophos was sprayed into tap water, and the obtained solution was concentrated to a specific volume and applied to RB-AuNPs. Table 3 shows that the ethoprophos recoveries were 102.4%, 105.4%, and 1.3.6%, based on the ratio of the absorption value at 600 nm to the absorption value at 520 nm. These findings confirmed that this method is applicable for detecting ethoprophos in real samples.

## 4. Conclusions

In this study, a simple colorimetric and fluorometric assay for detecting ethoprophos with high sensitivity and specificity using RB-AuNPs was established. The assay relied on ligand replacement on the surface of AuNPs, resulting in AuNP aggregation. This aggregation, in turn, permitted colorimetric visualization and fluorescence enhancement. The linear range of detection was calculated at the submicromolar level, and the LOD was determined to be 37.0 nM. This simple method was successfully applied to detect ethoprophos in tap water with high reliability. It is believed that this simple assay can be useful in monitoring ethoprophos on-site, especially in combination with other methods such as lab-on-chip techniques.

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
