# Peer review of "A Simple Colorimetric and Fluorescent Sensor to Detect Organophosphate Pesticides Based on Adenosine Triphosphate-Modified Gold Nanoparticles"

_sensors, 2018, doi:10.3390/s18124302_

Round 1
Reviewer 1 Report
This paper reports a sensor system for the detection of organophosphate pesticides, which takes the advantage of aggregation induced spectral changes in AuNPs. The authors also test the sensing capability of their system in water. I have several concerns regarding the presentation of data in this manuscript:
1. At what pH were these studies conducted? Did the authors try to maintain the pH, or even measure the pH changes? I would think that pH would have a major role in the sensing in such systems but author do not touch upon this aspect at all. No information has been provided.
2. Authors describe 3sigma/k as LOD, but do not explain the significance of this? What does this value represent?
3. What is the real significance of “detection value” that has been mentioned numerous times in the paper, if there is any? Is it just the read out value obtained?
4. For section 3.1 and 3.2 of results and discussion, I could not find critical details about the experiments. What was the concentration of pesticides used? Was only one concentration used? If so, on what basis was it selected? Did the authors do any competition experiments –if not then why?
5. Section 3.3 (lines 182-191) is a repetition of analysis presented in section 3.1 and 3.2. The auhors have already mentioned that RB-AuNPs were made using ATP. Effect of adding ATP as in RB-AuNPs should be discussed in section 3.2.
6. What concentrations of ATP were tested? How was the response for different tested concentrations? No data is presented.
7. The authors need to compare the performance of their sensor with other literature reported AuNP-based OP sensors.
Other comments:
8. Full names of OPs should be provided in Fig 1 alongside the abbreviations.
9. What is ROA (not explained anywhere – had to figure out myself).
10. Font size for axis labels of graphs (3b, 3d, 5a, 6c) should be increased.
Author Response
Response to Reviewer 1 Comments
Point 1: At what pH were these studies conducted? Did the authors try to maintain the pH, or even measure the pH changes? I would think that pH would have a major role in the sensing in such systems but author do not touch upon this aspect at all. No information has been provided.
Response 1: The effect of pH on the assay for EP had been conducted. The results indicated that this assay was applicable at range of pH from 3.0 to 10.0. These results were added in Figure 6 in the section 3.4.
Point 2: Authors describe 3sigma/k as LOD, but do not explain the significance of this? What does this value represent?
Response 2: 3σ/k as LOD is commonly accepted to determine LOD. σ: standard deviation of the 12 blank test; k: the slope of linear fitting curve. Please refer to the references (J. Anal. Toxicol. 2009, 33,129–142; Anal. Chem. 2011, 83, 813–819).
According to the IUPAC definition, when the signal-to-noise ratio equals 3, the signal is considered to be a true signal. Therefore, the LOD can be extrapolated from the linear calibration curve when the signal equals three times of the noise.
Point 3: What is the real significance of “detection value” that has been mentioned numerous times in the paper, if there is any? Is it just the read out value obtained?
Response 3: We are so sorry to missing the explanation of “detection value”. Detection value means the change of absorption at 520 nm without and with pesticides. The detection value was replaced by ΔA520 in the revised manuscript.
Point 4: For section 3.1 and 3.2 of results and discussion, I could not find critical details about the experiments. What was the concentration of pesticides used? Was only one concentration used? If so, on what basis was it selected? Did the authors do any competition experiments –if not then why?
Response 4: We are so sorry for it. We have already put the concentration of pesticides used. The competition experiments were conducted (Figure 3e). These changes have been highlighted in the revised manuscript.
Point 5: Section 3.3 (lines 182-191) is a repetition of analysis presented in section 3.1 and 3.2. The authors have already mentioned that RB-AuNPs were made using ATP. Effect of adding ATP as in RB-AuNPs should be discussed in section 3.2.
Response 5: This is a good suggestion. The effect of adding ATP as in RB-AuNPs on assay for pesticides was discussed in section 3.2.
Point 6: Section 3.3 (lines 182-191) is a repetition of analysis presented in section 3.1 and 3.2. The authors have already mentioned that RB-AuNPs were made using ATP. Effect of adding ATP as in RB-AuNPs should be discussed in section 3.2.
Response 6: This is a good question. We tested various concentrations of ATP (from 0.8 to 5.0 mM). The response to EP was in the range of 0.8 to 1.4 mM. However, the RB-AuNPs showed some response to PF in the low concentrations, and it took long time to assay EP with low sensitivity in the high concentrations. Given specificity and quickness of assay, 1.0-1.2 mM ATP was optimized as final concentration. The detailed data was summarized in table 1.
Point 7: The authors need to compare the performance of their sensor with other literature reported AuNP-based OP sensors.
Response 7: We have listed other literature reported AuNPs-based OP sensors in table 2. Most of them lack the selectivity for specific OP.
Point 8: Full names of OPs should be provided in Fig 1 alongside the abbreviations.
Response 8: We have put the full names of OPs in Figure 1.
Point 9: What is ROA (not explained anywhere – had to figure out myself).
Response 9: ROA is the ratio of absorption at 600 nm and 520 nm. We have put the explanation in the revised manuscript.
Point 10: Font size for axis labels of graphs (3b, 3d, 5a, 6c) should be increased.
Response 9: We have already increased the font size for axis labels of graphs (3b, 3d, 5a, 6c).
Reviewer 2 Report
It is an impressive way of detecting pesticides. However, I have some concerns before providing my comments.
Did author try any inteference study?
Is there any significant effect of lights as optical methods specially fluorescence is certainly affected by lights.
In my understanding, for such colorimetric assays, one should not go beyond absorption 1.0 A.U whereas authors tested it up to 2.00, please explain why did you think its needed.
Can you re-do the experiments keeping in mind the suggestions.
All figures should be replaced with higher resolution figures.
Author Response
Response to Reviewer 2 Comments
Point 1: It is an impressive way of detecting pesticides. However, I have some concerns before providing my comments.
Response 1: Thank reviewer for his/her good comments and suggestions for our manuscript.
Point 2: Did author try any interference study?
Response 2: We have done interference study at the range of quantitative detection, and the results indicated that interferences in the presence of other pesticides were negligible. These results were summarized in Figure 3e.
Point 3: Is there any significant effect of lights as optical methods specially fluorescence is certainly affected by lights.
Response 3: Not significant effect. Here, rhodamine B was used in this assay, and it is very stable against light. All of fluorescence experiments were done at max excitation wavelength of rhodamine B (520 nm). The sensitivity of fluorescence should decrease at other wavelength excitation of rhodamine B.
Point 4: In my understanding, for such colorimetric assays, one should not go beyond absorption 1.0 A.U whereas authors tested it up to 2.00, please explain why did you think its needed.
Response 4: We agreed with reviewer. Usually, the absorption of gold nanoparticles used for colorimetric sensor is below 1.0 A.U.. Here, we have also tried various concentration of gold nanoparticles, and the response to EP is optimal in high concentration.
Point 5: Can you re-do the experiments keeping in mind the suggestions.
Response 5: Yes, we have done the all the experiments according to your good suggestions.
Point 6: All figures should be replaced with higher resolution figures.
Response 6: We have already increased the resolution of figures.
Round 2
Reviewer 1 Report
The authors have made satisfactory changes to the MS based on earlier remarks. The MS can be accepted for publication in its current form.